# A Sentinel-2 Machine Learning Dataset for Tree Species Classification in Germany

Maximilian Freudenberg[1], Sebastian Schnell[2], and Paul Magdon[3]

[1]Chair of Forest Inventory and Remote Sensing & Neural Data Science Group, University of Göttingen, Germany
[2]Thünen Institute of Forest Ecosystems, Eberswalde, Germany
[3]Faculty of Resource Management, University of Applied Sciences and Arts (HAWK), Göttingen, Germany

**Abstract.** We present a machine learning dataset for tree species classification in Sentinel-2 satellite image time series of bottom of atmosphere reflectance. It is geared towards training classifiers, but less suitable for validating resulting maps. The dataset is based on the German national forest inventory of 2012, as well as analysis ready satellite imagery computed using the FORCE processing pipeline. From the national forest inventory data, we extracted the tree positions, filtered 387 775 trees in the upper canopy layer and automatically extracted the corresponding bottom of atmosphere reflectance time series from Sentinel-2 L2A images. These time series are labeled with the corresponding tree species, which allows pixel-wise classification tasks. Furthermore, we provide auxiliary information such as the approximate tree position, the year of possible disturbance events or the diameter at breast height. Temporally, the dataset spans the years from July 2015 to end of October 2022 with ca. 75.3 million data points for trees of 48 species and 3 species groups, as well as 13.8 million observations for non-tree background. Spatially, it covers entire Germany. The dataset is available under following DOI (Freudenberg et al., 2024): https://doi.org/10.3220/DATA20240402122351-0

# 1 Introduction

Climate change increases the risk of severe weather events such as heavy rainfall or droughts in Central Europe (Toreti et al., 2023). The recent past has seen large-scale forest diebacks due to drought, disease or insect manifestations or a combination of these disturbances (Senf et al., 2020; Senf and Seidl, 2021b). Forest managers face the challenge of adapting their management practices through diversification and other strategies to mitigate these threats. Here, remote sensing will play an increasingly important role as it can support well-informed decisions by providing extensive land cover and forest information at higher temporal frequencies than ground-based forest monitoring approaches. In this context, information on tree species is essential for many forest management decisions.

Tree species classification in satellite imagery is important, not only for scientific, but also for practical applications in forestry and nature conservation. This task has been in focus since the early days of space-borne remote sensing with the first Landsat sensors (Walsh, 1980) and it continues today with the application of machine-learning methods to large areas (Bolyn et al., 2022; Blickensdörfer et al., 2024).

Sentinel-2 (S2) satellite images are the ideal basis for such analyses, as they are standardized, freely available and collected with high temporal revisit frequency. Machine learning, particularly deep learning, is commonly employed to tackle classification tasks in image data, albeit requiring substantial amounts of training data (Bolyn et al., 2022; Lake et al., 2022; Yuan and Lin, 2020). Deep learning is a type of machine learning that uses neural networks with multiple layers to automatically learn patterns from large datasets (Goodfellow et al., 2016). In the context of tree species classification, generating training data is demanding and one has to resort to visual interpretation and on-screen labeling of high resolution aerial images, ideally combined with validation in the field – or one has to source labels from forest inventory data.

Ahlswede et al. (2023) have addressed the problem of training data compilation and created a multi-modal training dataset, containing aerial, as well as Sentinel-1 and 2 images of over 50 000 sites in the state of Lower Saxony, Germany. The dataset contains image-wise labels for 20 European tree species, generated from stand level forest inventory data. Utilizing different deep learning models, the authors achieved an $F_1$ score of 54.6%, using Sentinel-2 data alone. The $F_1$ score is the harmonic mean of user's and producer's accuracy, or precision and recall, respectively. They conclude that "the integration of multi-seasonal data might disentangle further species-related information regarding phenology phases" (Ahlswede et al., 2023, p. 691) – this is what we aim for with the dataset presented here.

Hemmerling et al. (2021) used exactly this kind of multi-seasonal Sentinel-2 data to classify 17 different tree species in the state of Brandenburg, Germany. They applied a random forest classifier to time series of the years 2018 and 2019 and reached $F_1$ scores between 67% and 99% for the nine most frequent species, thereby demonstrating that at least a subset of species can be separated using S2 time series comparable to the ones provided here. As in the first study, the authors obtained their labels from forest inventories conducted by state authorities.

These two studies are noteworthy exceptions regarding the amount of training data used, because the used datasets were relatively large. Fassnacht et al. (2016) reviewed studies on tree species classification from remotely sensed data and conclude that "*investigations focusing on [..] a single often comparably small test site by far dominated the reviewed studies*". This

hinders the generalizability of results and the applicability of generated models to other areas: a dataset covering a large area and long time spans is needed.

To overcome the problem of limited training data we tap the largest dataset of field observations of tree species in Germany: the national forest inventory (NFI). The German NFI runs on a cycle of 10 years, with a subsample after 5 years, and covers more than 25 000 sites, over 60 000 sampling points and more than 500 000 trees across all ownerships and site conditions (Polley et al., 2018). For each tree, several variables such as species, relative position and diameter at breast height (DBH, 1.3 m) are recorded. The resulting dataset is the most comprehensive available for German forests and the derived statistics provide valuable insights into the forest condition, composition and development on regional and national level. However, the design of the NFI was not tailored for creating remote sensing reference datasets but to provide an efficient sampling and plot design for estimating key forest variables. From a remote sensing perspective, one of the major caveats is, that the exact sampling positions need to be kept confidential, e.g., to prevent biased estimates when management practices are changed in the plot vicinity.

The goal of the work presented is twofold: first, to make satellite data at NFI plot positions available for third parties without revealing the exact geolocations and second, to analyze the separability and temporal patterns of tree crown reflectances for tree species in Germany. We link NFI records to bottom of atmosphere reflectance (BOA) time series from matching Sentinel-2 images, enabling tree species classification and other applications for a broad range of potential users. Said time series were extracted from analysis ready data generated by the Framework for Operational Radiometric Correction for Environmental monitoring (FORCE) (Frantz, 2019), hosted on the CODE-DE[1] platform. The resulting dataset provides BOA reflectances from July 2015 to October 2022 and in sum contains the time series of 387 775 individual trees and 70 242 non-tree locations. Multiplying the counts of tree and non-tree locations with their individual number of observed time steps yields a total of ca. 75.3 million data points for trees and 13.8 million observations for non-tree background, covering the entirety of Germany and 48 tree species and 3 species groups. The primary purpose of the data is training classifiers for detecting tree species in satellite images from the Sentinel-2 mission, but it could also be used for studying phenological and spectral patterns of tree species. It is less suitable for validating maps at pixel level. The dataset is available online under https://doi.org/10.3220/ DATA20240402122351-0 (Freudenberg et al., 2024) with CC BY 4.0 license.

## 2 Materials and methods

### 2.1 Study area and national forest inventory

The dataset covers the entire area of Germany, including islands. More specifically, it contains 24 925 of the 25 382 cluster plots recorded in the 2012 national forest inventory. The missing cluster plots either contained only trees below the canopy layer, the field inventory was conducted in a non-standard way (e.g. with custom post-processing of the coordinates) or the cluster plot coordinates were simply missing from the database we obtained. Temperate broadleaf and mixed forests prevail in most

---

[1]https://code-de.org

regions of the country. Coniferous forests, mainly consisting of *Picea abies* (Norway spruce), dominate at higher elevations and forests with *Pinus sylvestris* (Scots pine) occur on the sandy soils of the north-eastern part of the country. In 2022, about 32% of Germany was covered by forest (Riedel et al., 2024). Heavy droughts and following insect infestations in the years 2018–2022 resulted in a decline of growing stock in certain areas (Reinosch et al., 2024; Thonfeld et al., 2022; Holzwarth et al., 2023).

The German national forest inventory is conducted on a regular, square sampling grid as shown in Figure 1 with a grid size of 4 km × 4 km or less, depending on the federal state. At each grid point there are four inventory plots, aligned in a 150 m × 150 m square. The south-western corner of the square aligns with the 4 km × 4 km grid, as shown in Figure 2.

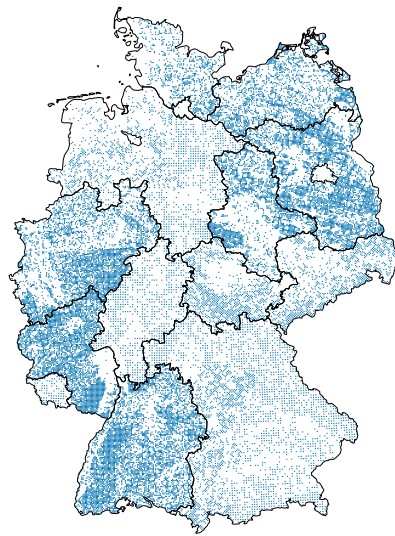

**Figure 1.** The sampling positions of the German national forest inventory 2012. Borders: © GeoBasis-DE / BKG (2024)

The geolocation of each subplot is measured with a Global Navigation Satellite System (GNSS) device, which may or may not be differentially corrected using correction information from terrestrial reference stations. At the subplot, two angle count samplings are performed (Gregoire and Valentine, 2007), which means that trees whose diameter at breast height (DBH) covers more than a certain solid angle are recorded.

The first angle count sampling includes all trees within a distance from the sample location of 25 times their DBH (basal area factor 4). The positions of the selected trees are determined by measuring their azimuth angle using a compass and their distance to the plot center by an ultrasonic device (Haglöf Vertex or similar) or in edge cases via measuring tape. Furthermore, the tree species, DBH and other variables are recorded. At these measured tree positions, the BOA reflectances were extracted and related to the corresponding, ground-measured information - how this was done is described later.

A second angle count sampling captures the surrounding forest composition by recording the species of all trees within a radius of 33.34 or 50 times their DBH (basal area factor 2 or 1), depending on how many trees were observed in the first sampling with basal area factor 4. The second angle count sampling allows to tell, which sub-plots are pure stands, i.e. have

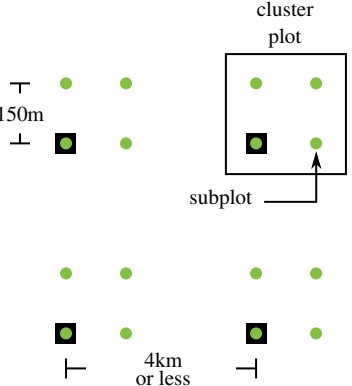

**Figure 2.** The German national forest inventory sampling grid (black squares) and the subplots (green). The south-western subplot in each cluster plot is aligned with the overarching grid.

only one tree species in them. The information about stand purity is included in the dataset, so that the end user can filter for trees in pure stands.

## 2.2 NFI reference data selection

The reflectances recorded in a Sentinel-2 pixel represent the mixture of all land cover – or in our case tree species – within the pixel. However, in closed canopy forests the BOA reflectance is dominated by the uppermost canopy layer and we can safely assume that trees overshadowed by larger individuals contribute only little to the overall reflectance within a pixel. To compile the provided training dataset we therefore filtered the NFI data for trees that are probably visible from above. We first removed all trees that grow in the understory; this information is recorded during the inventory. For the remaining trees we modeled a

circular growing space using the NFI's official method described in (Riedel et al., 2017, pp. 39, 40). The model establishes a species-specific linear relation between basal area and the growing space of a tree. The growing space is a measure for the area occupied by a tree as a whole in an idealized forest stand. In lack of a model for direct estimation of crown area from basal area, we use the growing space as a proxy for it and stick to the term "crown area". The model is defined in equation A1 and the parameters are supplied in Table A1 in the appendix. As we know the position of each tree, as well as its predicted crown area,

we removed trees that are probably not visible from above by a heuristic algorithm. We counted a tree as visible if it either had the largest basal area of all trees within a radius of 3 m or if its crown area was overlapped by less than 50% by surrounding trees. Trees classified as visible by this heuristic formed the basis for the training dataset.

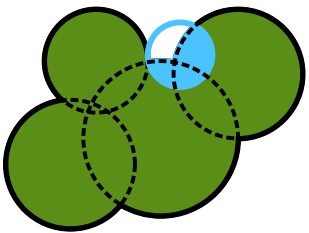

**Figure 3.** Sketch of a tree group: Green trees are assumed to be visible. The blue tree overlaps with more than 50% of its area with other trees and is therefore discarded.

To allow training classification methods for the discrimination between tree and non-tree pixels, we added non-forest observations to the dataset. For this, we sampled the tree cover density layer provided by the Copernicus Land Monitoring Service for the year 2018 within a 300 m × 300 m patch around the NFI plots[2]. The tree cover density layer is sampled at locations that are at least 20 meters away from the next pixel with tree density greater than 10%.

### 2.3 Satellite data selection

We used images from the Sentinel-2 satellites, pre-processed to analysis-ready level, which includes atmospheric correction and cloud masking, by the FORCE processing pipeline (Frantz, 2019). FORCE provides a way to compute harmonized time series that are spatially and spectrally well aligned, which is discussed in more detail later. The resulting data comprises all S2 bands with 10 or 20 m resolution, with the 20 m bands pan-sharpened (resampled) to 10 m resolution. Additionally, FORCE provides quality assurance information (QAI) that aids in filtering out undesirable image conditions such as clouds, snow, or high water vapor content. The data is hosted on the CODE-DE[3] and EO-Lab[4] platforms. End users have the option to either download the pre-processed data or can re-process it using the same settings utilized in generating the FORCE data cube on CODE-DE. The necessary parameter files are provided alongside the dataset.

### 2.4 Time series extraction and data processing

Previous studies have taken different approaches for linking forest field data with satellite images: Many work only with pure stands (e.g., Verhulst et al. (2024); Hościło and Lewandowska (2019)), while others assign plot-specific species compositions (e.g., Blickensdörfer et al. (2024)). Some sample individual pixels within polygons (Hemmerling et al., 2021; Grabska-Szwagrzyk et al., 2024), cut out pixels covered by a fixed area plot (Persson et al., 2018), or even calculate reflectances at the single tree crown level (Plakman et al., 2020).

As the German NFI performs angle count sampling, it is not possible to exactly determine how much of a given area (e.g. a Sentinel-2 pixel) is covered by which tree species and adjacent land cover types. In such cases, there are mainly two approaches, one can follow for relating field and satellite data, which we call "tree-centric" and "pixel-centric". The "tree-centric" approach

---

[2]https://land.copernicus.eu/en/products/high-resolution-layer-tree-cover-density

[3]https://code-de.org

[4]https://eo-lab.org

assigns the most probable reflectance value to individual tree crowns by directly extracting them from the satellite image. In contrast, the "pixel-centric" approach labels individual pixels with species information, for example, a species composition derived from inventory data. We chose the tree-centric approach, as it accounts best for the response design of the angle count sampling. With angle count sampling, an assignment of species information to pixels is difficult because it does not provide a full census of all trees over a specifiable area.

We started by clipping 300 m × 300 m image patches containing the 24 925 filtered NFI cluster plots and their surroundings from the FORCE data cube, as depicted in Figure 4. We extracted the bottom of atmosphere reflectance (BOA) as well as the quality assurance information (QAI). Before extraction, we filtered the plots to ensure they contained at least one pixel with data, not affected by clouds or cloud shadows.

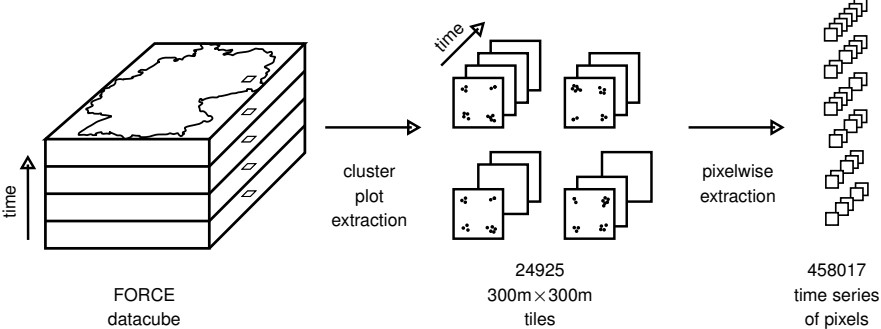

**Figure 4.** The time series extraction workflow: First, 300 m × 300 m tiles are clipped from the FORCE datacube for Germany for all records between 2012 and 2022. Second, the pixel-wise time series are extracted from the tile time series.

In a second step, we extracted the BOA and QAI pixel time series from the extracted patches at each tree position. In cases where a single tree covered more than one 10 m × 10 m Sentinel pixel, we calculated the area-weighted average of all pixels intersected by the tree's crown area, as depicted in Figure 5. Each extracted satellite observation was then linked to its acquisition date, the corresponding NFI data and more information. Senf and Seidl (2021a) provide a Landsat-based map of forest disturbances for Germany between 1986 and 2020 at a resolution of 30 m. To be able to identify possible disturbance events, we included the disturbance year from this map in the dataset. However, this still leaves a gap between 2020 and 2022, for which no disturbance information is available. This was bridged by attaching the information whether the trees were still present during the 2022 NFI. To enable approximate spatial analyses, we furthermore included the center coordinate of the 1 km INSPIRE grid tile the cluster plots are located in. The INSPIRE grids (INSPIRE MIG, 2023) are a set of Pan-European geographical grid systems in the ETRS89-LAEA coordinate reference system with their origin at 52° N 10 ° E. The grids have a power-of-ten spacing in meters; we used the 1 km grid.

The final dataset comprises the columns presented in Table 1 and an excerpt is given in Table C1 in the appendix. All samples were randomly split into training and validation sets based on their cluster plot IDs with a ratio of 70% - 30%. This rules out

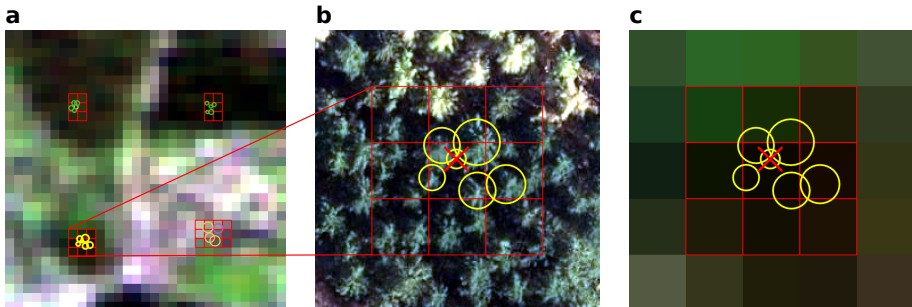

**Figure 5.** (a) The whole cluster plot cutout of 300 m×300 m. S2-Image: European Space Agency (2021) (b) The lower left subplot with the corresponding orthophoto for reference. Douglas firs in the lower part, Norway spruce in the upper part of the image. Image: © BKG (2021) (c) The S2 pixels corresponding to the subplot with circles depicting the modeled tree crown areas. The crossed out tree is omitted because it overlaps too much with surrounding trees.

any spatial overlap between the training and test sets and reduces correlations between the two. For benchmark studies, we recommend using this split to ensure comparability across publications.

## 2.5 Assessment of the geolocation accuracy of the NFI plots

The tree positions in the NFI are measured in polar coordinates relative to the plot center, using a compass for the angle and an ultrasonic device for the distance measurement. We assume that the errors for angle and distance are small compared to the GNSS error of the plot center position measurement. GNSS measurements can be differentially corrected by using ground-based reference stations to increase positional accuracy. Depending on the federal state and field team, coordinates of the plot centers are measured with corrected GNSS devices or not. Of the sub-plots with trees in the dataset, 76.5% were corrected,

22.5% were not, and the remainder has unknown status.

To estimate the accuracy of the plot center coordinates, we compared the field-measured tree positions with tree positions derived from true-ortho aerial images, obtained from the Federal Agency for Cartography and Geodesy. These images are ortho-rectified using a surface model and aligned with high accuracy to ground control points. The ATKIS orthophoto standard guarantees a geolocation error with standard deviation of 0.4 m or less (Arbeitsgemeinschaft der Vermessungsverwaltungen

der Länder der Bundesrepublik Deutschland (AdV), 2020). Two expert image interpreters then manually shifted a sample of 200 NFI plot positions, and thereby the trees, to match the true tree positions by comparing local tree patterns as depicted in Figure 6. This allows to quantitatively evaluate the deviation of measured from true positions and to compare the accuracy of corrected and uncorrected measurements.

---

[5]https://force-eo.readthedocs.io/en/latest/howto/qai.html#quality-bits-in-force

**Table 1.** Dataset contents and column description.

| Column name | Data type | Description |
| --- | --- | --- |
| tree_id | Integer | A globally unique tree id. Negative values represent non-tree records. |
| tnr | Integer | Cluster plot id |
| enr | Integer | Corner id (1-4). Negative values represent non-tree records. |
| time | Integer | The acquisition date, encoded as Unix time, representing the number of seconds elapsed since 1970-01-01, 00:00 UTC. Every date was randomly shifted by up to three days. |
| species | Integer | The tree species, encoded according to the official NFI schema, provided within the dataset in a separate table "x_ba". |
| boa | byte array | The BOA reflectance values: 10 signed 16-bit integers, one for each band, encoded as 20 byte blob. To hamper the identification of exact plot positions, each value was multiplied with a uniform random number between 0.95 and 1.05. |
| qai | Integer | Quality assurance information bit-flags, encoded as 16-bit integers, allowing for filtering based on image quality. The FORCE documentation provides details on the meaning of each bit[5]. |
| is_train | Bool | Whether the record belongs to the training or validation set. |
| is_pure | Bool | Whether the record comes from a pure stand according to the NFI definition. |
| dbh_mm | Integer | Diameter at breast height (1.3 m) in millimeters. |
| height_dm | Integer | Tree height in decimeters. |
| crown_area_m2 | Float | Modeled tree crown area in $m^2$. |
| x_wgs84 | Float | Longitude of the corresponding 1 km Inspire grid tile center. |
| y_wgs84 | Float | Latitude of the corresponding 1 km Inspire grid tile center. |
| is_corrected | Bool | Whether the NFI position measurement was differentially corrected. |
| disturbance_year | Integer | The disturbance year according to the map provided by Senf and Seidl (2021a). |
| present_2022 | Bool | Whether the tree was observed again in the 2022 forest inventory. |
| doy | Integer | The day of year of the acquisition, corresponding to the shifted date. |

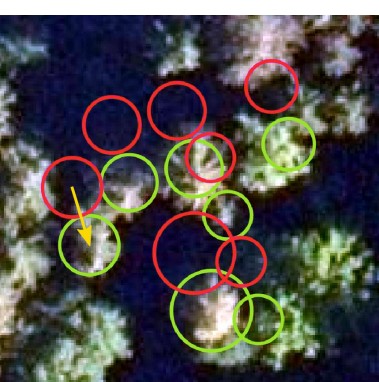

**Figure 6.** Original, measured GNSS coordinates (red) were shifted (here by 4.8 m) to the visually best matching position (green) in aerial orthophotos to quantify GNSS errors. Circles depict modeled crown areas.

## 2.6 Species separability analysis

To detect inconsistencies within the dataset, we computed the infrared reflectance histograms of five species for mixed and pure stands. If the histogram shows artifacts like double peaks or differs strongly between pure and mixed stands, this could hint to deficiencies in the respective part of the dataset. The histograms were computed for band B8 (842 nm), averaged over all records in June 2021 for a sample of five species whose occurrence is correlated – *Betula pendula* often grows along with *Pinus sylvestris* and *Fagus sylvatica* often appears together with *Quercus* spp. June 2021 has been chosen because both Sentinel satellites were operational and, unlike the preceding years and 2022, 2021 was not particularly dry.

## 3 Dataset description and statistics

### 3.1 Numerical species distribution

Due to the highly varying dominance of tree species in Germany, the numerical distribution of the different species (Figure 7) is heavily imbalanced. The most abundant species is *Pinus sylvestris* (Scots pine), followed by *Picea abies* (Norway spruce), *Fagus sylvatica* (European beech) and the different *Quercus* (Oak) species. A complete list of included tree species and their counts can be found in appendix Table C3.

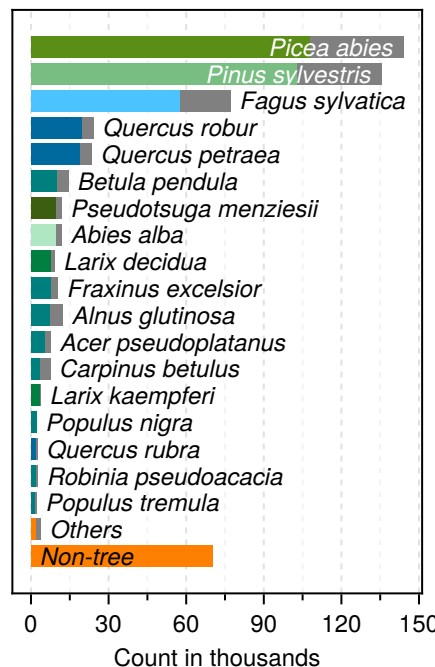

**Figure 7.** The numerical species distribution in the training dataset (colored) and in the original NFI 2012 data (gray).

## 3.2   Temporal signatures of selected species

Evergreen and deciduous trees can be clearly separated visually by inspecting the time series of their infrared (IR) reflectance, as depicted in Figure 8. In the presented time series, the observations for a given species and point in time have been averaged across all undisturbed individuals in pure stands. Whether a stand is pure or not was determined using the second angle count sampling of the NFI (basal area factor of 1 or 2). Obviously, deciduous broadleaf trees exhibit a much stronger seasonal pattern than evergreen coniferous trees in our dataset. This separation is less evident in the green band, likely due to its higher susceptibility to atmospheric effects and its lower absolute reflectance, which combine to diminish the signal to noise ratio. While the temporal infrared profiles of *Fagus sylvatica* and *Quercus robur* are generally distinguishable across most years, there are instances where differentiation becomes challenging (e.g. 2016 and 2020). *Quercus robur* tends to have a slightly lower IR reflectance on average, particularly in summer. *Picea abies* and *Pinus sylvestris* also differ only slightly in the infrared, with *Picea abies* having lower average values on trend. Overall, differentiating species by their temporal profiles alone seems challenging without considering their spectrum at the same time. Figure B1 in the appendix depicts the same data as Figure 8 but additionally includes error bands that were omitted here for clarity.

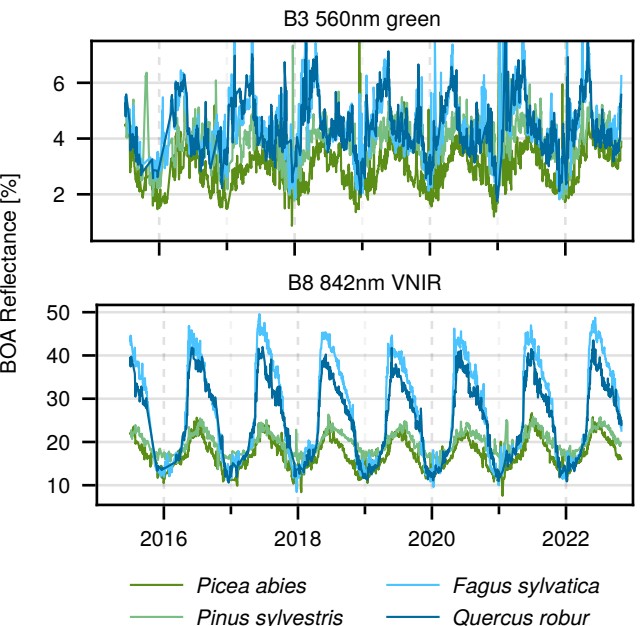

**Figure 8.** Time series of BOA reflectance for indicated species, averaged over all undisturbed individual trees in pure stands at a given time. The data has been filtered to exclude all types of cloud cover and their shadows, snow, and pixels with high aerosol optical depth.

Looking at a random selection of four individual trees' time series, depicted in Figure 9, it becomes clear that at the level of a single tree, the differences between species still seem to be present, but with high variance from year to year.

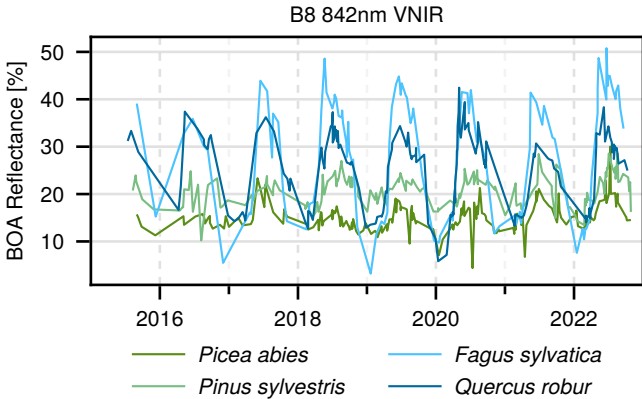

**Figure 9.** Time series of random single trees of different species.

Figure 10 shows the total observation count over time, i.e. how often each tree was imaged within a month, summed up across all trees. After the commissioning of Sentinel-2B in June 2017 the number of observations increases. As one would expect, there are more observations in the summer months when clouds are less likely and especially from 2018 onward the counts regularly reach over 1 million.

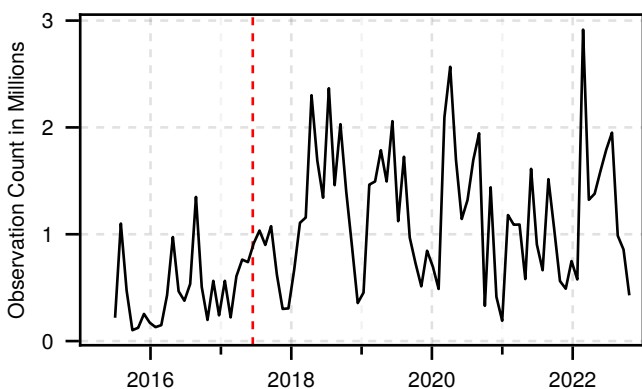

**Figure 10.** Total monthly observations of all trees in the dataset (tree count multiplied by individual observation count per month). The vertical red line corresponds to the Sentinel-2B commissioning date.

### 3.3 Spectral signatures

Besides the temporal variation of the reflectance, the spectral variation is an important feature for the tree species classification – however, the species are not necessarily separable by their spectrum alone, as can be seen in Figure 11. It depicts the Sentinel-2 spectra of the five most frequent species, as well as the background class. *Fagus sylvatica* and *Quercus petraea* for example have almost matching spectra, especially in the shorter wavelengths. The resulting spectra match the ones presented in Immitzer et al. (2016).

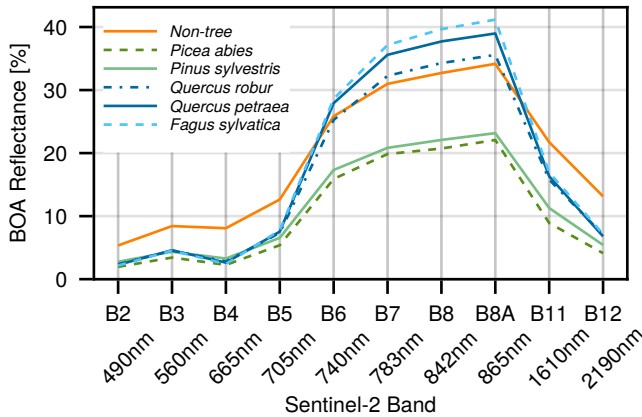

**Figure 11.** Average spectrum of the five most frequent species in the dataset, plus the background class. Records from pure stands have been averaged between May and August (inclusive) of the years 2017–2022.

### 3.4 Spatial distribution

It can be expected that the temporal signatures vary with local conditions, e.g. along an latitudinal or elevation gradient. Therefore, it is important to analyze the spatial coverage of the training data. Figure 12 shows that *Picea abies* (a) is mainly present in the south-west of Germany and in the lower mountain ranges. *Pinus sylvestris* (b) on the other hand, is predominant on the sandy soils of the north-eastern part of the country. The different *Quercus* species (c) occur mostly in the west of Germany, but are also present throughout the rest of the country. *Fagus sylvatica* (d), lastly, co-occurs with *Quercus* spp., but in contrast to them, manages to settle in the higher and therefore colder hillscapes of the central parts of Germany. Note however, that these spatial distributions are derived from the dataset, which does not mirror the NFI one to one due to filtering and the availability of satellite images.

### 3.5 NFI geolocation accuracy estimation

The analysis of the spatial accuracy of the NFI plot coordinate GNSS measurements reveals that 95% of corrected GNSS positions deviated by less than 11.2 m, and 81% by less than 5 m; Figure 13 depicts the corresponding histogram along with the empirical cumulative density function. Against expectations, the comparison of corrected and uncorrected GNSS measurements shows no significant difference.

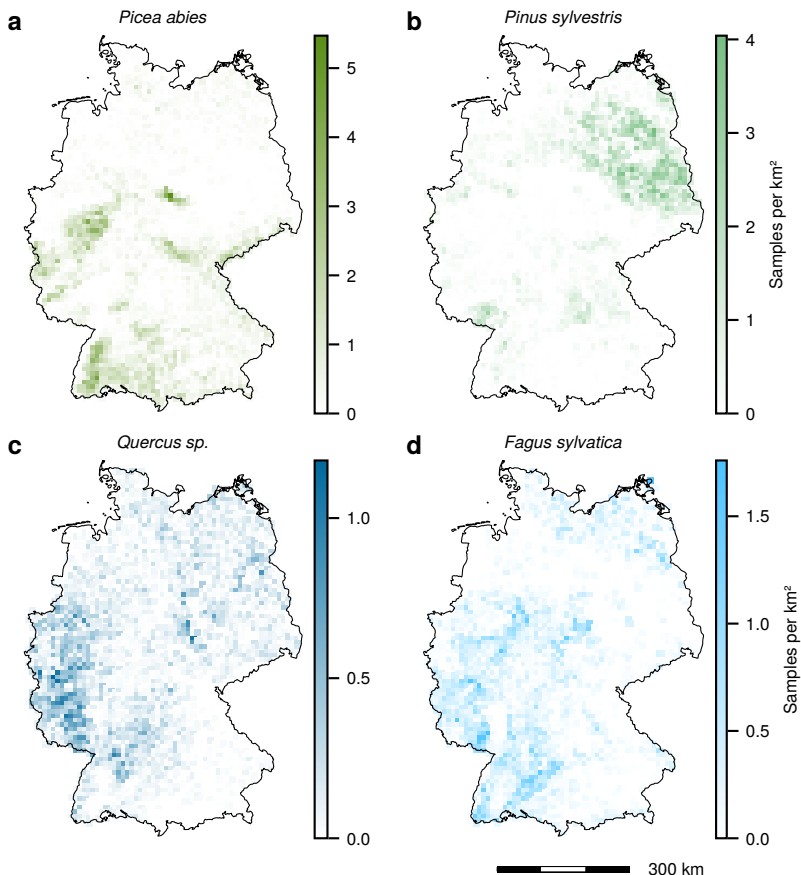

**Figure 12.** Spatial tree distribution for different species. Note the different scales. Borders: © GeoBasis-DE / BKG (2024)

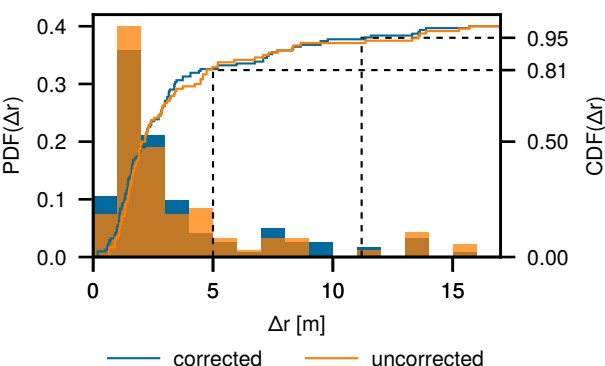

**Figure 13.** Histogram of distances by which plot locations were shifted from the original GNSS positions. Differentially corrected measurements are depicted in blue.

## 3.6 Separability analysis

Figure 14 shows the histograms of S2 band B8 (842nm) averaged over all records in June 2021 for the species pairs *Betula pendula – Pinus sylvestris* and *Fagus sylvatica – Quercus robur – Quercus petraea*, each computed over mixed and pure stands, respectively. These combinations were chosen because the respective species are often co-occurring. The reflectance distributions for *Pinus* and *Betula* clearly differ between mixed and pure stands. In mixed stands, the distributions are relatively wide and overlap, whereas in pure stands, there are separable peaks (albeit some overlap remains) and the distance between maxima is larger. Comparing *Fagus sylvatica* to the two *Quercus* species, one can see that the distributions overlap much more, as all three species are broad-leaved. In mixed stands, there is hardly any observable difference between the distributions. For pure stands, the distributions still overlap significantly, but the distance between peaks is slightly larger than in mixed stands.

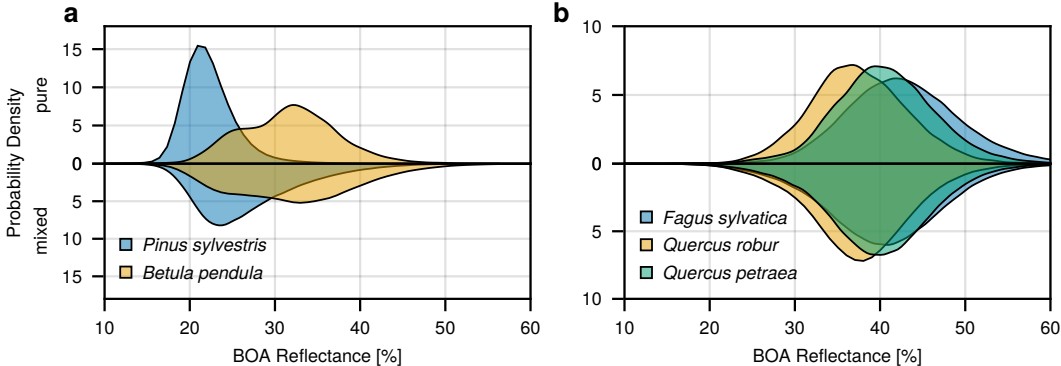

**Figure 14.** Histogram of near infrared (842 nm) BOA reflectances, averaged over all trees in June 2021, for (a) *Pinus sylvestris* and *Betula pendula* and (b) *Fagus sylvatica*, *Quercus robur* and *Quercus petraea*. The upper parts represent pure stands and the lower parts mixed stands.

## 4 Discussion

### 4.1 Geolocation accuracy

**Sentinel-2:** To obtain the presented dataset, we linked spatial information from two different data sources: georeferenced satellite images and on-ground GNSS measurements. A misalignment of these sources might lead to extracting wrong pixel values from the image data. FORCE co-registers all Sentinel-2 images with averaged Landsat time series. The Landsat images are in turn co-registered with the Sentinel-2 global reference image which results in a geometric accuracy of 10.2 m at the 90% confidence level for Landsat 8 (Haque et al., 2022) (8 m at 80% confidence). Consequently, this is the best estimate for the spatial accuracy of the used S2 images. The reason for this cyclic co-registration of Sentinel to Landsat to Sentinel is, that so far only the S2 level 1 archive has been processed to a common standard[6]. The level 2 data, which compensates atmospheric effects and is needed for coherent time series, is not yet available at a standardized processing baseline in any public archive.

---

[6]https://sentinels.copernicus.eu/web/sentinel/technical-guides/sentinel-2-msi/copernicus-sentinel-2-collection-1-availability-status

**NFI geolocation accuracy:** The comparison of corrected and uncorrected GNSS measurements showed no significant difference in spatial accuracy, at least not the way we measured it. As differential correction unquestionably increases the GNSS accuracy, we suppose that increasing the count of sampled plots as well as the number of image interpreters would change our result. Furthermore, trees growing skew and outliers when matching the crown patterns might have negatively influenced the results. Lastly, it will be interesting to analyze the accuracy of trained classifiers as a function of correction status.

**Combined geolocation accuracy:** The combined geolocation accuracy is difficult to compute for several reasons: 1) the satellite images are corrected by FORCE, as discussed above, 2) the satellite image accuracy is latitude- and time-dependent[7] and 3) the GNSS errors we measured do not follow a Gaussian distribution. Neglecting these points and using the values derived for the 80% confidence level, namely 8 m for the satellite images and 5 m for the GNSS positions, we obtain an error estimate of 9.4 m. This is nearly equivalent to the pixel size, which means that the extracted pixel values are still likely to represent a reasonable approximation of the targeted trees, whose diameter is of comparable size. Lastly, the error of 9.4 m is likely overestimated for two reasons: first, the true error distributions of GNSS and satellite geolocation error magnitudes is Log-Normal (opposed to a Rayleigh distribution), typically having a higher share of small magnitudes. Second, the 8 m satellite geolocation error is derived for a single Sentinel-2 scene, but it is averaged out by the co-registration process.

## 4.2 Adverse imaging conditions

During the extraction process, we filtered out most pixels with cloud cover or cloud shadows. FORCE employs the FMASK algorithm (Zhu and Woodcock, 2012) for cloud detection, which has an accuracy of 84% for cloud / clear detection and 72% detection accuracy for cloud shadows (Aybar et al., 2022). Consequently, falsely labeled image regions lead to commission or omission errors in the final dataset, i.e. usable pixels might have been removed by being labeled as cloudy or cloud pixels could be in the dataset. However, there are other imaging conditions that might affect the quality of a pixel like high aerosol content, snow or poor illumination conditions. FORCE encodes this information in the quality assurance information and end users can use this to further narrow the dataset down to only the highest quality pixels.

## 4.3 Extraction of non-forest points

The non-forest points were randomly sampled within the extracted 300 m × 300 m tiles. In consequence, we only sampled non-forest points from areas like city centers or industrial zones where they are situated close to forest – which is rather unlikely. Therefore the extracted non-forest points are biased towards rural villages and agricultural areas.

## 4.4 Taxonomic identification

The field teams of the NFI data are trained and undergo testing before being allowed to take samples. However, it cannot be ruled out that under adverse conditions certain species are confused. We cannot quantify this error, but assume that the vast majority of tree species identifications are correct, in particular for the common species.

---

[7]S2 Data Quality Reports: https://sentiwiki.copernicus.eu/web/document-library

## 4.5 Tree-centric pixel extraction

As described in section 2.4, there are different ways of linking satellite image data with field information. We took a "tree-centric" approach and extracted reflectances for every tree in the dataset. Compared to the pixel-centric approach, this comes with advantages and disadvantages. The main advantage of the tree-centric approach is that it reflects the response design of angle count sampling and one obtains the best estimate for the spectral reflectance of a given tree crown. This allows to extract data also for rare species and such, that only appear in mixed stands, albeit their statistics will be influenced by mixed pixels. Lastly, one does not need to define a dominant species based on arbitrary thresholds, as done in Persson et al. (2018), for example.

A drawback in comparison to the pixel-centric approach is, that different tree crowns can receive the same spectral signature. For example, if two tree crowns are completely located within the same S2 pixel, they receive identical values and information is duplicated. We checked the non-randomized dataset for duplicate bottom of atmosphere reflectances among the tree records. Non-tree points were sampled from a larger area, so duplication plays no role in their case. To identify duplicates, we grouped the dataset by cluster id, corner id, time and reflectance spectrum. If there were N identical reflectances per group, we counted N-1 as duplicates. In total, the dataset subset for trees contains ca. 4.87 million duplicate entries out of ca. 66 million, which translates to 7.38%. Out of these 4.87 M duplicates, 3.86 M (5.84%) are duplicates with identical species label and 1.01 M (1.53%) have differing species labels. Ergo, *at least* 0.77% (1.01 M / 66 M / 2) of the labels are wrong.

Should the user wish to reduce the correlation between samples or remove duplicate pixel time series, we recommend the following procedure: first, group the dataset by subplot; second, compute the correlation of the full time series between the different trees in the plot; and finally, remove all trees that correlate beyond a certain threshold, except for one.

A weakness that both approaches, tree-centric and pixel-centric, share are mixed pixels: At present, we cannot exactly quantify the effect of pixels that contain different tree species on our dataset, as it is in most cases impossible to derive the species shares of a pixel based on the NFI data. The NFI does not fully sample a given plot, so in most cases, labels are only available for parts of a given pixel. Another source for mixed pixels are the 20 m resolution bands of Sentinel-2 that are pan-sharpened to 10 m by FORCE, thereby distributing identical information across several pixels.

## 4.6 Species separability analysis

Figure 14 (a) showed that the IR-reflectance distributions of *Pinus* and *Betula* are wide and overlap in mixed stands, whereas they are more separated in pure stands. We interpret this as a potential indication that, at least for this species pair, the dataset may contain mislabeled data due to insufficient spatial accuracy or that the extracted pixel values originate from mixed pixels containing other species or land cover classes.

In contrast, comparing *Fagus* and *Quercus* spp. in mixed and pure stands revealed no significant differences, with the reflectance distributions overlapping substantially. However, this does not necessarily indicate labeling errors; it could also reflect naturally occurring values. This highlights the necessity of including factors beyond spectral data, e.g. temporal profiles as shown in Figure 9, for accurate species classification.

### 4.7 Considerations for map production

The purpose of this dataset is to train classifiers, ultimately for mapping of tree species. These classifiers will have a certain model accuracy, derived from e.g. the validation split of the presented dataset. However, caution is required when judging the accuracy of generated maps based on the model accuracy - the model accuracy should not be used as sole basis for validation. This applies to the current dataset as well, since heuristic data filtering or pixel duplication may have altered the distribution of reflectance values in a way that could negatively affect the results. Users should also consider this when applying additional data filters, e.g. for the diameter at breast height. Consequently, we recommend auxiliary data for validating generated maps. If users wish to validate maps based on the presented dataset, they can do so using aggregate statistics, such as at state level, since the dataset contains coarse location data that makes it suitable for such analyses. However, such analyses will be restricted to trees that are visible from above according to our heuristic. Further, areas covered by tree species, as derived from produced maps, can be compared to population estimates from the NFI, which are publicly available (J.H. von Thünen-Institut, 2024). Lastly, the "area of applicability" approach developed by Meyer and Pebesma (2021) can be used to ensure that models are not applied outside the predictor variable space, thus preventing potential bias.

## 5 Conclusion and outlook

In this work, we present the most comprehensive dataset so far of annotated Sentinel-2 time series data for tree species detection in Germany. With over 380 thousand trees of 48 species observed for over seven years, this dataset can significantly advance research into automatic tree species classification for Germany, and central Europe. At the same time the described approach can serve as a pilot study for making national forest inventory data from other countries accessible for the remote sensing community e.g. for training machine learning models without releasing the exact geolocations publicly. Lessons learned from its application can be used to enhance future inventories and datasets. For example, it could show that for underrepresented species more labels are required, which in turn could be sampled in targeted inventories.

As discussed in the previous section, the dataset still has several shortcomings that could be improved. To achieve better agreement between labels and images, the spatial accuracy of the data sources has to be increased. To do so, we suggest that in future all NFI position measurements are taken using differential GNSS devices, although we saw no significant differences in accuracy. Furthermore, we expect that aligning the Sentinel-2 images directly with the S2 global reference image instead of averaged Landsat time series would improve their spatial accuracy and make it easier to derive interpretable error metrics. We consider releasing an updated dataset version as soon as Sentinel-2 L2A collection one is fully accessible.

The main focus of further efforts will be to increase the number of labels for weakly represented classes, e.g. by utilizing automatically classified high resolution orthophotos as reference. First attempts to automatically identify underrepresented tree species in standard RGBI aerial images with 20 cm spatial resolution have failed, so the presented dataset is still limited regarding less abundant species. Another option to increase the overall amount of data would be to incorporate forest inventory data at the stand level from e.g. state forest enterprises, however, this data often only provides estimates of tree species proportions within management units, but no geolocation of individuals.

We hope that this dataset fosters the research into time series-based classification of tree species and believe it offers many possibilities for analyses that go beyond the ones presented here. Users can freely recombine the provided data and for example calculate basal or crown area proportions per sampling location and use this information as labels instead. Using classification methods in general, one could investigate which spectral bands and which points in time are crucial for precise species classification. As the dataset not only contains the time series of individual trees' BOA reflectances, but also their approximate location, spatio-temporal patterns in tree phenology could be assessed on individual species level. For example, the onset of leaf emergence could be analyzed first in the dataset alone, and later by using species maps generated by a derived classification method. Lastly, the dataset could be used to correlate reflectances and approximate health conditions with meteorological events like droughts on a per-species level. This would open up further research into climate-change resistant species and enables the identification of endangered forest stands. In the future we plan to release updated versions of the dataset, particularly after the final publication of the 2022 NFI.

## 6 Data availability

All data is available online under https://doi.org/10.3220/DATA20240402122351-0 (Freudenberg et al., 2024) with CC BY 4.0 license.

*Author contributions.* MF: coding, assembling the dataset, main work on manuscript, SS: data provision, proof reading, advice in research questions, PM: advice in research questions, manuscript development, proof reading

*Competing interests.* The authors declare to have no conflicts of interest.

*Acknowledgements.* The authors thank the Thünen Institute of Forest Ecosystems for providing the national forest inventory data. MF thanks Alexander Ecker for ongoing financial support and reviews, as well as Christoph Kleinn and Ryan Carroll for proof-reading.

*Financial support.* The *Klimba* project and this work were funded by the Federal Ministry for Digital and Transport under grant number 50EW2012A/B.

## Appendix A: Crown area estimation

Following equation is used to model the crown area, using the parameters $\alpha$ and $\beta$ from table A1 (Riedel et al., 2017, pp. 39, 40).

$$A_C = \alpha + \beta A_B \tag{A1}$$

$A_C$: Tree crown area, $A_B$: Basal area

**Table A1.** Parameters of the crown area equation (Riedel et al., 2017, p. 40). We corrected the $\alpha$ value for poplar; the original value is 23, which is a typing error.

| Tree Species Group | $\alpha$ [m²] | $\beta$ | Max. Stem Count | Assigned Tree Species |
|---|---|---|---|---|
| Fir | 2.85 | 200 | 3500 | All firs except hemlock |
| Douglas Fir | 5.00 | 200 | 2000 | Douglas fir |
| Pine | 1.00 | 300 | 10000 | All pines |
| European Larch | 5.00 | 285 | 2000 | European larch |
| Japanese Larch | 5.00 | 260 | 2000 | Japanese larch (+ hybrids) |
| Beech | 1.33 | 300 | 7500 | Beech, hornbeam (whitebeam) |
| Oak | 1.11 | 395 | 9000 | Pedunculate oak, sessile oak, Turkey oak, swamp oak |
| Red Oak | 2.50 | 350 | 4000 | Red oak |
| Ash | 2.50 | 330 | 4000 | All other deciduous trees not mentioned |
| Alder | 2.50 | 435 | 4000 | Alder, black alder, white alder/grey alder, green alder |
| Birch | 2.50 | 525 | 4000 | Silver birch, downy birch (+ Carpathian birch) |
| Poplar | 2.30 | 320 | 350 | All poplars |
| Spruce | 2.85 | 195 | 3500 | All spruces as well as arborvitae, hemlock, sequoia, yew, Lawson cypress, other conifers |

## Appendix B: Additional Figures

## Appendix C: Database excerpt and species counts

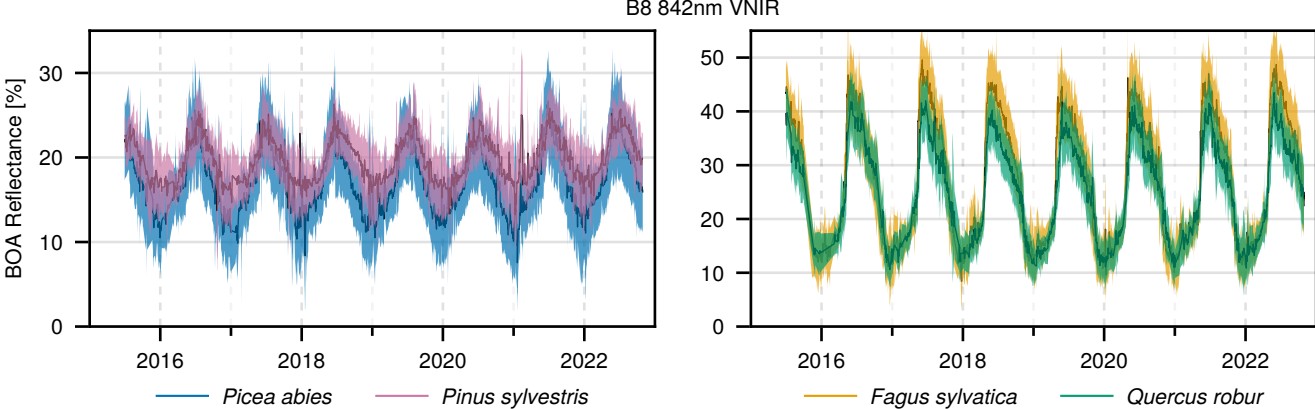

**Figure B1.** Time series of infrared reflectance and the standard deviation for indicated species, averaged over all undisturbed individual trees in pure stands at a given time. The bands have a width of 2 standard deviations. The data has been filtered to exclude all types of cloud cover and their shadows, snow, and pixels with high aerosol optical depth.

**Table C1.** Database excerpt. The bottom of atmosphere (BOA) reflectance is encoded as 10 signed 16 bit integers, the quality assurance information (QAI) is a single 16 bit integer. DOY abbreviates day of year.

| tnr (cluster id) | enr (corner id) | tree_id | species | time | BOA | QAI | is_train | is_pure | |
|---|---|---|---|---|---|---|---|---|---|
| 455 | 1 | 69831 | 211 | 1440374400 | 10 16-bit integers | 8192 | 1 | 0 | ... |
| 455 | 1 | 69831 | 211 | 1448064000 | 10 16-bit integers | 10256 | 1 | 0 | ... |
| 455 | 1 | 69831 | 211 | 1455494400 | 10 16-bit integers | 10240 | 1 | 0 | ... |
| 455 | 1 | 69831 | 211 | 1460592000 | 10 16-bit integers | 8192 | 1 | 0 | ... |
| 455 | 1 | 69831 | 211 | 1463961600 | 10 16-bit integers | 8192 | 1 | 0 | ... |
| 455 | 1 | 69831 | 211 | 1467072000 | 10 16-bit integers | 8192 | 1 | 0 | ... |
| ⋮ | ⋮ | ⋮ | ⋮ | ⋮ | ⋮ | ⋮ | ⋮ | ⋮ | ⋱ |

| | dbh_mm | height_dm | crown_area_m2 | x_wgs84 | y_wgs84 | is_corrected | disturbance_year | present_ 2022 | DOY |
|---|---|---|---|---|---|---|---|---|---|
| ... | 231 | 243 | 20.4 | 9.80714 | 47.64294 | 1 | 0 | 1 | 236 |
| ... | 231 | 243 | 20.4 | 9.80714 | 47.64294 | 1 | 0 | 1 | 325 |
| ... | 231 | 243 | 20.4 | 9.80714 | 47.64294 | 1 | 0 | 1 | 46 |
| ... | 231 | 243 | 20.4 | 9.80714 | 47.64294 | 1 | 0 | 1 | 105 |
| ... | 231 | 243 | 20.4 | 9.80714 | 47.64294 | 1 | 0 | 1 | 144 |
| ... | 231 | 243 | 20.4 | 9.80714 | 47.64294 | 1 | 0 | 1 | 180 |
| ... | ⋮ | ⋮ | ⋮ | ⋮ | ⋮ | ⋮ | ⋮ | ⋮ | ⋮ |

**Table C2.** List of all included tree species with counts (part 1).

| Species Code | Species | | Common Name | Count |
|---|---|---|---|---|
| -1 | - | - | other land cover | 70242 |
| 10 | *Picea* | *abies* | Norway spruce | 107798 |
| 12 | *Picea* | *sitchensis* | sitka spruce | 937 |
| 19 | *Picea* | spec. | other spruces | 232 |
| 20 | *Pinus* | *sylvestris* | Scots pine | 102730 |
| 21 | *Pinus* | *mugo* | mountain pine | 88 |
| 22 | *Pinus* | *nigra* | European black pine | 606 |
| 24 | *Pinus* | *cembra* | Swiss pine | 3 |
| 25 | *Pinus* | *strobus* | eastern white pine | 431 |
| 29 | *Pinus* | spec. | other pines | 65 |
| 30 | *Abies* | *alba* | silver fir | 9375 |
| 33 | *Abies* | *grandis* | grand fir | 384 |
| 39 | *Abies* | spec. | other firs | 291 |
| 40 | *Pseudotsuga* | *menziesii* | Douglas fir | 9598 |
| 50 | *Larix* | *decidua* | European larch | 7674 |
| 51 | *Larix* | *kaempferi* | Japanese larch (+hybrids) | 3308 |
| 90 | | | other coniferous trees | 139 |
| 94 | *Taxus* | *baccata* | European yew | 11 |
| 100 | *Fagus* | *sylvatica* | beech | 57341 |
| 110 | *Quercus* | *robur* | English oak | 19617 |
| 111 | *Quercus* | *petraea* | sessile oak | 18697 |
| 112 | *Quercus* | *rubra* | Northern red oak | 1861 |
| 120 | *Fraxinus* | *excelsior* | common ash | 7469 |
| 130 | *Carpinus* | *betulus* | hornbeam | 3411 |
| 140 | *Acer* | *pseudoplatanus* | sycamore maple | 5042 |
| 141 | *Acer* | *platanoides* | Norway maple | 598 |
| 142 | *Acer* | *campestre* | field maple | 387 |
| 150 | *Tilia* | spec. | linden tree (indigenous species) | 1294 |
| 160 | *Robinia* | *pseudoacacia* | black locust | 1553 |
| 170 | *Ulmus* | spec. | elm, native species | 406 |
| 181 | *Castanea* | *sativa* | chestnut | 416 |
| 190 | | | misc. broadleaf trees with long life expectancy | 246 |
| 191 | *Sorbus* | *domestica* | service tree | 2 |
| 193 | *Sorbus* | *aria* | common whitebeam | 51 |
| 200 | *Betula* | *pendula* | silver birch | 9729 |
| 201 | *Betula* | *pubescens* | moor birch | 858 |
| 211 | *Alnus* | *glutinosa* | black alder | 7098 |
| 212 | *Alnus* | *incana* | grey alder | 460 |
| 220 | *Populus* | *tremula* | common aspen | 1402 |
| 221 | *Populus* | *nigra* | European black poplar (+ hybrids) | 1945 |

**Table C3.** List of all included tree species with counts (part 2).

| Species Code | Species | | Common Name | Count |
|---:|---|---|---|---:|
| 222 | *Populus* | *x canescens* | grey poplar (+hybrids) | 196 |
| 223 | *Populus* | *alba* | silver poplar | 109 |
| 224 | *Populus* | *trichocarpa x maximoviczii* | balsam poplar | 636 |
| 230 | *Sorbus* | *aucuparia* | European rowan | 270 |
| 240 | *Salix* | spec. | willow | 1203 |
| 250 | *Prunus* | *padus* | bird cherry | 77 |
| 251 | *Prunus* | *avium* | wild cherry | 1357 |
| 252 | *Prunus* | *serotina* | black cherry | 132 |
| 290 | | | misc. broadleaf trees with short life expectancy | 92 |
| 292 | *Malus* | *sylvestris* | European crab apple | 37 |
| 293 | *Pyrus* | *communis* | European wild pear | 42 |
| 295 | *Sorbus* | *torminalis* | wild service tree | 71 |

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
