# Peer review of "A Sentinel-2 Machine Learning Dataset for Tree Species Classification in Germany"

_Earth System Science Data, 2024_

## Author Comment (AC1)

**Response to First Review Comments**

2024-09-05

Dear Reviewers, Dear Editor,

We would like to thank the editorial team of ESSD for handling our manuscript and the swift replies to requests. Equally, we thank the anonymous reviewers for their critical and constructive questions they have raised with respect to the manuscript "A Sentinel-2 Machine Learning Dataset for Tree Species Classification in Germany". We highly appreciate that you invested your time and effort, which is not self-evident. The valuable comments helped us to improve the manuscript substantially.
Below we have formulated our responses to the individual points raised by the reviewers. The updated manuscripts, as well as a document highlighting the differences have been uploaded.

On behalf of all authors,

Max Freudenberg

**1 Reviewer 1**

**1.1 Major comments**

1. First, the clarity of the general text can be improved. It was not entirely clear to me how the GNNS locations were used, how the exact locations of the individual trees were determined inside the plots and how their crown area was calculated. I was also confused about the coordinates provided in the dataset, called the "Inspire-grid" coordinates, are these the plot centers? I think the clarity could be improved if the authors are more clearly stating what exactly the dataset represents. It is my understanding that the dataset represents plot data of pure stands of tree species and their spectral data from Sentinel-2 and not individual trees, but please correct me if I am wrong!

   Response: We indeed extracted reflectances on a single tree level, not aggregated values for pure stands. To address the questions, we improved the explanation how the tree location is measured within the plot (line 85), explained more clearly how the crown area was determined (line 98) and provided additional information on the "inspire grid" is (line 141).

2. Second, I noticed that throughout the text and in the figures and tables, the authors are not using the standard species naming guidelines. It is my understanding that scientific species names should be written in italics with genus name capitalized and the species name not.

   Response: We changed all occurrences of botanic species names in text, tables and figures accordingly.

3. Finally, the figures generally look great but the time series- and spectral signature plots (Figure 8 to 11) could be enlarged, there is enough space to enlarge the plots and it would make it easier for the reader to interpret the results. Furthermore, since Figure 8 and 9 depict averages of species, it could be a nice to show some variability around these averages in the form of shading or otherwise.

   Response: According to the journal style the final layout will have two columns. For this reason we matched the figure sizes that should appear within one column to the column's width. However, we now slightly enlarged them by trimming white space. We also produced a two-column-wide version of Figure 8, so that the editor can choose what fits the layout best. To visualize the standard deviations of the time series, we added Figure B1 to the appendix and referenced it in the main text.

**1.2 Minor comments**

1. L16 "disturbances" instead of "factors"?

   Response: Changed to "disturbances".

2. L33 please shortly explain or define the F1 score.

Response: We added a sentence with the definition (harmonic mean between precision and recall) (line 31).

3. L61-L64 please explain how data on 387,775 individual trees and 70,242 non-tree locations result in "75.3 million data points for trees and 13.8 million observations for non-tree background". Do these numbers refer to the number of images in the time series multiplied by the number of locations? This is presently unclear.

Response: We modified the sentence and now state that this number is obtained by multiplying the number of trees and non-tree location with their individual number of observed time steps in the satellite time series (line 62).

4. L64 "51 tree species and species groups" please clarify how many species and how many groups exactly, it could be 2 species and 49 groups, right?

Response: We added the clarification that there are 48 species and 3 species groups in the dataset to the abstract and introduction. (lines 8 & 64)

5. L68 "it contains 24 925 of the 25 382 cluster plots" what happened to the 467 plots not included? Please explain why these were not included in the analysis.

Response: We excluded plots that contained only trees below the canopy layer or plots where the field inventory was carried out in a non-standard way (e.g. the sampling positions were south-west instead of north-east from the tract location or GNSS positions underwent custom post-processing). We added this information to the text under "Study area and national forest inventory" (line 69).

6. L70 I assume scientific species names should be in italics "Pinus sylvestris" also please use the full English name of the species "Scots pine" and "Norway spruce" or synonyms.

Response: We switched all occurrences of botanic names to italics and adapted the common names as requested.

7. L73 Could give some references about forest disturbances in Germany since 2018 after "forest has likely decreased" such as: https://doi.org/10.1093/forestry/cpae038 and https://doi.org/10.3390/rs15174234

Response: We added three citations, including the suggested ones and clarified that the growing stock has decreased, not the forest area. The affected areas almost always remain forest (line 75).

8. L92 "we can remove trees that are probably not visible from above by a heuristic." A heuristic what? Function? Argument"?

Response: We changed this to "a heuristic algorithm" (line 104).

9. L94 "the biggest (area-wise)" what is meant by area-wise? Crown area or basal area or something else?

Response: We changed this to "in terms of basal area" (line 105).

10. Figure 5 shows polygon circles representing "modelled tree crowns" but I cannot find in the text how these were modelled. Please explain in detail how this was done because it is a critical part of the analysis. Was crown area measured in the field for each tree? If yes, how was this measured?

Response: The crown width is not measured in the field by the NFI. Therefore, we modeled the space occupied by a tree, its growing space, according to the official species-specific empirical NFI models (see Riedel, 2017, pp. 39-40), using the parameters and models described therein. According to Riedel et al. (2017) the growing space "corresponds approximately to the crown projection area" (our translation). We added two explanatory sentences to the section "NFI reference data selection" (line 98), clarifying what exactly was modeled, and added the equation and parameter table to the appendix.

11. L127 "Every date was randomly shifted by up to three days." Why was this done?

Response: This was done to reduce the risk of reverse-engineering of the exact NFI plot locations.

12. L146 Please specify the brand and model of the ultrasonic device.

Response: Distances are measured using the Haglöf Vertex 3 or 4 ultrasonic device. In edge cases, distances are measured using a measuring tape. This information was also added to the text (line 86).

13. L162-163 Please adapt to the species naming guidelines of the journal (I assume species scientific names should be in italics)

Response: Done.

14. Figure 7 might be a nice addition to show two panels: a) species distribution of all trees in the NFI and b) species distribution of trees extracted from the NFI. Also in this figure, please use italics for scientific species names.

    Response: We combined both distributions in one plot and now present the original distribution in the background.

15. L166 "coniferous vs deciduous" is not a useful distinction, you can have coniferous trees that are also deciduous (e.g. Larix). Deciduous says something about the leaf phenology while coniferous says something about the phylogeny (conifers being a subset of gymnosperms) which are not mutual exclusive or useful groups to make in this study. It would be much more useful to refer to "broadleaved deciduous" and "evergreen needleleaf" or in the case of larch to "deciduous needleleaf". Furthermore, the common holly (Ilex aquifolium) can also grow to tree size but is an "broadleaved evergreen" tree/shrub. Could be nice to add some columns to the dataset providing data on the leaf phenology (deciduous/evergreen), leaf shape (needle/broadleaf) and phylogeny (e.g. plant family).

    Response: We changed all occurrences of "deciduous" to "broadleaf". However, we did not add a new column to the dataset, as this can easily be done by the data users according to their specific needs. All trees with a species code below 100 are coniferous, which we expect is enough destinction for a start.

16. L179 "Figure 10 shows the total observation count over time." Make clear what observations these are, images, pixels, trees?

    Response: The observations refer to trees. The figure displays how often a tree was observed by a satellite image within a month, summed up across all trees. We added an explanatory sentence (line 190) and adapted the figure caption.

17. L190-L195 the text and the panels in Figure 12 are not in the same order, please change the order in either the text or the figure and label the panels in Figure 12 (a,b,c,d) and refer to the panels in the text.

    Response: We changed the order of figures and added sub-labels that are referenced in the text.

18. Table A3 & A4 Please adapt to the species naming guidelines of the journal (I assume species scientific names should be in italics)

    Response: Done.

**2 Reviewer 2**

**2.1 Major comments**

1. The dataset does not adhere to current practices for pixel-level training data. Typically, individual pixels or pixel blocks are selected and labeled based on the dominant tree species, forest type, or proportions of tree species (based on basal area). This method has been used in previous studies mapping tree species (as cited by the authors) and it is common practice when mapping land cover as well. The authors do not follow this approach. Instead, they select trees from the forest inventory and then extract Sentinel-2 pixels corresponding to each tree. Since trees are much smaller than the 10x10 and 20x20 meter Sentinel-2 pixels, this results in many duplicate pixels in the sample. In homogeneous field plots, pixel values and labels are replicated because many trees occupy the same pixel. In mixed species plots, pixels are replicated, and the same value is associated with different tree species. I have not encountered a study using such a sample for tree species classification, and the authors do not demonstrate the utility of this dataset. I believe pixel replication can potentially bias model training and error estimation. Therefore, I strongly suggest the authors follow current practices. For example, the authors could provide for each subplot: average reflectance, basal area proportions by tree species, crown area proportions by tree species, and/or other tree statistics.

    Response: We acknowledge that the common approach typically involves labeling individual pixels or pixel blocks based on dominant species or proportions. The plot design of the German NFI uses Bitterlich sampling (angle count sampling), which does not select all trees in a defined area (such as a Sentinel-2 pixel). Thus, assigning a species label to a fixed area using Bitterlich sampling data would also result in inaccuracies. We tried to use aerial images to derive a full area classification, however the image quality in Germany is not yet at a level that allows accurate enough detection. The only exemption could be single-species plots, where one can define a radius within which all trees belong to the same species with certainty. In the sampling design suggested by the reviewer, one would assign a species composition to one or more pixel(s) and train a classifier to predict exactly this composition. In our design, the classifier would predict the probability that a certain pixel is of a certain species, which is a different

information. But we expect that it is closely related and finally yields the same results if the number of training instances is large enough. We agree that our approach can generate duplicate pixels. However, we have now checked the dataset for duplicates in the bottom of atmosphere reflectances of the tree records. We grouped the dataset by tract id, corner id, time and reflectance values. If there were N identical reflectances per group, we counted N-1 as duplicates. In total, the dataset contains ca. 4.87 million duplicate entries in the tree data out of ca. 66 million, which is 7.38%. Out of these 4.87 M duplicates, 3.86 M (5.84%) were duplicates with identical species label and 1.01 M (1.53%) had differing species labels. Ergo, at least 0.77% (1.01 M / 66 M / 2) of the labels are erroneous. While having duplicates in the dataset is far from ideal, we expect this amount to be low enough as not to play a significant role when fitting machine learning models. For comparison: The widely used ImageNet dataset contains 14.2 M images, of which 1.2 M (8.5%) are duplicates and the LAION 400M dataset contains 60 M duplicated images (15%). To increase the transparency we extended the section "Mixed and duplicate pixels" in the discussion (line 265) to make this issue clear to the users. We also added methodological clarifications in the section "Time series extraction and data processing" (line 123) which now starts with a paragraph that contrasts the "traditional" approach with the one we chose and the conclusions now contain a sentence encouraging the user to recombine the data in new ways as appropriate to their needs (line 305). The German NFI was originally not designed as a reference dataset for remote sensing-based classifications but to provide statistically sound estimates efficiently. Despite deviations from common practices, the presented dataset still remains the largest single-tree dataset available in Germany. Therefore, we think that our reference dataset, even if not structured according to the common approach, is an important and valuable contribution to the remote sensing community.

2. The authors add random noise to the pixel reflectance to make the field plot locations untraceable. I understand that. However, the authors do not show a sensitivity analysis of the effect of adding noise. I would encourage the authors to test this effect on mapping accuracy. More generally, the authors could demonstrate the utility of their dataset for mapping tree species. The separability analysis is not really useful for map developers.

Response: The scope of this manuscript is to present the dataset itself and not the derived products, like a tree species map. There are many options how it could be used for map creation and the publication will empower the users to develop their own methods and research. Regarding the added noise: This is a requirement by the data owner who will not release the exact coordinates of the NFI plots. This is common practice for NFIs in Europe. However, we tested the performance of neural networks trained on noisy data, applied to non-randomized data. We conducted three training runs on the published dataset and an internal, "clean" version, classifying 13 tree species. The average accuracy for the runs on the noisy data was 76.1% and for the "clean" data 76%. In both cases, all training runs reached accuracies of 76+-0.3% (min/max). Therefore, we couldn't observe a drop in accuracy by adding 5% noise. But lastly, these values can only serve as rough estimate, as the actual susceptibility to noise depends on the model used. Regarding the separability analysis: this part might not be useful for map developers, but presents another aspect of the data (the reflectance distribution), so we kept it. However, we refactored the analysis as requested and split it into methods, results and discussion.

3. The writing could be improved with careful editing.

Response: We carefully reviewed and improved the text as described in the detailed responses to the reviewer comments. We also corrected typos and other errors.

**2.2 Minor comments**

1. L12: Avoid single-sentence paragraphs. Remove. Readers already got that information from the abstract.

Response: We removed the first sentence.

2. L19: traditional? Do you mean field information? I wouldn't term field inventories traditional. Both are needed.

Response: We changed "traditional [...] approaches" to "ground-based [...] approaches" (line 151).

3. L23: Extensive use?

Response: Changed "extensive" to "application [...] to large areas" (line 21).

4. L62: It is not possible to measure the reflectance of individual trees with Sentinel-2 data due to the spatial resolution. Sentinel-2 pixels represent a mixture of multiple tree canopies and background reflectance.

Response: We refer to our answer to the comment on L115 below.

5. L73: Area of stocked forest → forested area (as opposed to forest area)

   Response: Changed as requested.

6. L81: Please also specify the basal area factors associated with these radii.

   Response: We added that these radii correspond to BAF 4, 2 and 1 (lines 84, 90).

7. Fig 1. A figure of the sampling density by federal state would be more helpful. It is not possible to see the grid anyway

   Response: We added the state borders to the map, but did not switch to presenting the sampling density. Depicting the points directly allows the reader to locate high density areas, e.g. the Harz mountains.

8. L85: Please clarify what you mean with "subset of tree species labels". Is a label referring to pixels overlaying the angle count plot with BAF 4? Also, be specific about what you did and what can be done, e.g., "This information was used to label..." rather than "This... allows..."

   Response: A "label" refers to the information attached to pixel values that were extracted for each individual tree. We now state that the information on stand purity is included in the database, so it *can* be used to filter based on that criterion. We removed the sentence quoted last (line 92).

9. L90: stand area: you mean crown area?

   Response: We actually modeled the growing space. According to the source of the used model (Riedel et al., 2017) the growing space "corresponds approximately to the crown projection area" (our translation), so there should be no significant differences. We added a clarifying sentence (line 100).

10. L90: It is unclear (at this point) why you remove trees. In the previous section, you describe that you identify single-species stands. If you are removing trees in mixed stands, I wonder if your method tends to underestimate conifer trees, since their crown area is usually smaller and the crowns of the surrounding broadleaf trees are more flexible.

    Response: We now start the section "NFI reference data selection" (line 94) by explaining why we remove trees. Regarding the underestimation of conifer trees: We updated Figure 7 to also show the original NFI counts. In most cases, around 75% of the trees are retained and we do not notice a preference for broadleaf trees.

11. L92: same here. Write "..we removed trees..." rather than "..we can remove trees.."

    Response: Changed as requested.

12. L93: Use past tense consistently to describe what you did.

    Response: Changed as requested.

13. L96: Doesn't the forest inventory sampling design include non-forest observations? There are advantages with staying within a single sampling design.

    Response: The NFI does not visit non-forest sample points in the field. In principle we could stay within the same grid and sample from non-forest locations but initially we had only data for the patches with forest plots available. Overall there would be ca 190,000 cluster plots, which would increase the time it takes to extract the data and the storage requirements. We thus opted to sample in the extracted 300m x 300m patches. This could bias the non-tree samples towards agricultural land. We added a discussion section regarding this possible bias (line 252).

14. L96: comma in front of ", we added.."

    Response: Changed as requested.

15. L114: I am still confused by your wording. It sounds like you are extracting pixels associated with individual trees. You probably mean "when the projected tree crowns from the plot covered more than a single Sentinel-2 pixel,..."? Please clarify what your spatial unit in the field is. It is probably the subplot and not individual trees, i.e., you obtain a single reflectance measure for each subplot and S2 image.

    Response: We are indeed extracting the time series for each tree and changed the sentence from "extracted ... at the respective reference data position" to "extracted ... at each tree position" to make that clearer (line 133).

16. L115: Fair enough, but you are trying to be more precise than the data, considering that Sentinel-2 also has 20-m bands and the geolocation accuracy of the NFI and Sentinel-2.

    Response: We understand your concern and yes, this is at the edge of what is currently possible. For exactly this reason we are using the FORCE processing pipeline that aligns the S2 images with Landsat time series and thereby improves their geolocation accuracy. The approximate geolocation error of the satellite images is 8m at the 80% confidence level and the GNSS error amounts to 5m at the same confidence level. Computing the correct combined error is not trivial, as at least the GNSS error distribution is Non-Gaussian. Assuming Gaussian distribution, the combined error amounts to 9.4m. This is below the 10m pixel size, so extracted values should reasonably represent the targeted trees. We added a new paragraph to the discussion, treating the combined error (line 238). With the publication of Sentinel-2 Collection 1 (`https://sentinels.copernicus.eu/web/sentinel/sentinel-data-access/sentinel-products/sentinel-2-data-products/collection-1-level-2a`) the spatial accuracy of the source images will be further improved. The latest data quality report (`https://sentiwiki.copernicus.eu/__attachments/1673423/OMPC.CS.DQR.001.06-2024%20-%20MSI%20L1C%20DQR%20July%202024%20-%20101.0.pdf?inst-v=7071b3cb-abd4-4add-b2c2-d6e89a0d956a`) mentions an absolute geolocation accuracy of 8.56m for S2-A and 8.18m for S2-B at the 95% confidence level. We consider using S2 Collection 1 images for future database updates.

17. L124: Ok. You do seem to extract S2 reflectance values for each tree. What is the rational and use case behind it? An S2 pixel is a mixture of different surface types including different trees, understory and other background. For tree species mapping, we are usually interested in modeling the relationship between such mixed signal either with the dominant tree species, a forest type category, or fractions of trees species. Attempting to link individual trees with Sentinel-2 is uncommon. As a result, you will produce multiple replicates of the same pixel or 3x3 pixel group and associate them with multiple trees of the same or multiple species.

    Response: We kindly refer to our response to major comment one.

18. L125: I recommend to document your data in a table format. A table could show the field name of in your dataset along with a description.

    Response: Changed as requested. We replaced the enumeration by a table.

19. L142: Because you replicate pixels of subplot, your data still contains auto-correlated samples. There is also spatial autocorrelation within cluster plots. Do you have a recommendation how users can deal with the autocorrelation associated with cluster plots?

    Response: Yes, the autocorrelation increases with decreasing distance and positions within a subplot are close to each other. This is a general property of the underlying satellite data, independent from the method used to extract samples / pixels. So the method we used for extracting the pixels amplifies the already existing autocorrelation level. One option to deal with correlated samples would be to group the data by subplot, compute the correlation (e.g. using the time series), and remove samples that correlate beyond a certain threshold. This should effectively remove duplicated samples. We added this suggestion to the discussion section (line 272).

20. Figure 7: Why does the x-axis (count) only go up to 110? How does this correspond to your 350,000 trees and/or 25,000 cluster plots?

    Response: The axes represents thousands, which we depicted by "k". We switched the axis label to "count in thousands" and slightly altered the plot on request of reviewer 1.

21. Figure 12: Please replace "Samples per 100km2" with a more meaningful unit, e.g. trees per ha or tree species proportion.

    Response: We switched to samples per $km^2$.

22. 4.5 This section reports new results but is presented in the Discussion section.

    Response: We restructured the section into Methods - Results - Discussion.

---

## Referee Report (RR1)

**Review of Freudenberg, Schnell and Magdon "A Sentinel-2 Machine Learning Dataset for Tree Species Classification in Germany"**

Freudenberg, Schnell and Magdon have carefully revised their manuscript, taking into account my comments and suggestions and those of the other reviewer. I still believe this dataset is unique in its size and spatial extent and therefore potentially very useful for scientists, forest managers and policy makers alike. I appreciate the addition of the separability analysis in the results section. Besides some minor changes in the wording of some sentences (see minor comments) I have no further objections to the publication of this data paper. The paper would benefit from a thorough copy editing and typesetting revision by the journal to improve clarity and correct language use.

**Minor comments**

L26-L29 "Machine learning, particularly deep learning [...]" As I understand it, deep learning is a form of machine learning that uses many data layers and artificial neural networks in classification tasks, often applied in image recognition. It is also a buzzword. Please add references to this sentence to studies in which deep learning was used for tree species classification or similar. Good to shortly explain the difference between deep learning and machine learning here.

L77 "but due" indicates a decline in forest area from the 32% in 2012, but a decline in growing stock does not necessarily mean a decline in forest area, for example thinning. Please rephrase.

L109 "The growing space "approximately corresponds to the crown projection area"(Riedel et al., 2017, pp. 39, author's translation), so we use these terms interchangeably in the following" No this should not be used interchangeably because it is very confusing to the reader. "approximately corresponds" is not sufficient to use the terms interchangeably. The "growing space" here is not defined, neither are its units. Is it cubic meter? This should really be changed, be careful with the use of terms and their units.

L113-L114 still very unclearly written, not sure how trees were selected as visible. Please rephrase.

L196 "Obviously, broadleaf trees exhibit a much stronger seasonal pattern", not if they are evergreen broadleaf trees... please separate leaf shape (broadleaf or needleleaf) and phenology (evergreen and deciduous) more clearly. For example, holly (*Ilex*) is an evergreen broadleaf tree/shrub in Europe that does not likely show an obvious seasonal pattern in reflectance. On the other hand, Larch (*Larix*) is a common deciduous coniferous needleleaf tree. Suggested edit: "Obviously, deciduous broadleaf trees exhibit a much stronger seasonal pattern than the evergreen coniferous trees in our dataset."

---

## Referee Report (RR2)

**Major comments:**

the authors cover many important topics regarding reference data for large area mapping of tree species and supply a valuable dataset for the research community.
While the dataset will lack behind expectation with regard to scientific freedom and flexibility, its provision will provide researchers, educators and students with additional data for the investigation of important research questions related to climate change, forest preservation, forest management and biodiversity studies.
While some aspects of the data set, such as preprocessing, could be up for discussion, its publication will serve as a baseline for future publications of state and federal data sets and hopefully motivate more government authorities to provide their inventory data to the public.
The writing style is excellent throughout the paper with only a single recommendation from my side.
The researchers worked thoroughly on assuring high data quality and investigated the data set at hand for important characteristics, such as distinguishability of species from spectral signatures and geolocation accuracy, something that is to be expected from future related research.
In my opinion, a few aspects of possible data usage were missed in the study design and discussion but overall, the state of the art in the field of tree species mapping with multi spectral satellite imagery is presented correctly.

**Minor comments:**

106-107: "if their crown is overlapped by less than 50% by the surrounding trees" would be clearer language IMO

2.3:
Additional TSA-processing withing the FORCE framework is not possible and the opportunity to create a dataset of even higher quality is missed. This would undoubtedly lower the amount of available tree observations but might help classification approaches that are sensitive to noise from cloud shadows and fog.

134-135: calculating the area-weighted average of a pixels might be a big source of noise if, let's say, the other 75% of that pixel depict a substantially different type of land cover than the target tree species. Think of the spectral signature of a deciduous tree that is added to a coniferous evergreen and the undergrowth signal in winter observations.
In my opinion, some sort of outlier detection should be put in place to detect possible addition of noise.

276: This might be due to *Pinus'* often very top-heavy crown in plantations that allows undergrowth to be more visible. In combination with *Betulas* characteristic bark, it is no wonder that the signal gets mixed up. There might be similar issues with stands including *Larix* or *Fraxinus*.

310: One additional idea for use of your dataset could be the investigation of mixed pixels.
For large area mapping it would be great to know if any given mixture of species within a single pixel can be learned by a classifier.

**On the discussion of pixel level vs tree level reference data:**

While I see possible issues with area-weighted pixel extraction, I do not agree with the criticism stated by Reviewer 2:

FORCE uses the ImproPhe algorithm (Frantz 2016) that alters the pixel values of the 20m bands. To my understanding, duplicate values within the vicinity of a datapoint and thus spatial autocorrelation will be quite unlikely given the large size of the dataset. I can also support the author's claim, that duplicates (as well as random noise) within a certain threshold as stated in regard to the LAION 400M dataset are no issues for modern machine learning algorithms, especially neural networks, from my personal experience.

The addition of random noise is a valid point of criticism. However, as long as European NFI rely on fixed position sample plots, this approach seems to be the only viable method to provide data to the research community.

---

## Author Response (AR2)

**Response to Second Round of Review Comments**

2024-11-23

Dear Editor,

thank you for giving us the opportunity to resubmit our revised manuscript titled *"A Sentinel-2 Machine Learning Dataset for Tree Species Classification in Germany"* to ESSD. We genuinely appreciate the time and effort you and the reviewers have devoted for offering valuable feedback on our manuscript.

We are grateful to the reviewers for their insightful comments and suggestions, which have significantly improved the quality of our work. One of the major points raised by reviewer 2 concerned the method we used to assign labels in the training dataset, leading to an engaging and productive discussion about the pros and cons of various labeling strategies.

To address this, we have revised portions of the paper by introducing a new terminology that we define as "tree-centric" and "pixel-centric" labeling strategies. In the updated manuscript, these terms are introduced in Section 2.4. This section was partly rewritten and now contains a short literature review, demonstrating the lack of a "standard approach" and highlighting the variety of strategies that have been published for assigning tree species labels from forest inventory or management data. We further explore the advantages and disadvantages of the tree-centric pixel extraction approach for dataset usage in Section 4.5 ("Tree-centric pixel extraction") and discuss the implications and limitations ("Dos" and "Don'ts") for map production in the newly written Section 4.7, titled "Considerations for map production". Furthermore, the text has been reviewed by a native speaker familiar with remote sensing.

As the first freely accessible dataset derived from the German NFI for tree species classification, we are confident it will be well-received by researchers in the remote sensing community. Thank you once again for your consideration, and we look forward to your response.

Below, we have formulated our responses to the individual points raised by the reviewers. The updated manuscript, as well as a document highlighting the differences, have been uploaded.

On behalf of all authors,

Max Freudenberg

**1 Reviewer 1**

**1.1 Minor comments**

1. L26-L29 "Machine learning, particularly deep learning [. . . ]" As I understand it, deep learning is a form of machine learning that uses many data layers and artificial neural networks in classification tasks, often applied in image recognition. It is also a buzzword. Please add references to this sentence to studies in which deep learning was used for tree species classification or similar. Good to shortly explain the difference between deep learning and machine learning here.

   Response: We added three citations and a short explanatory sentence regarding deep learning (line 27).

2. L77 "but due" indicates a decline in forest area from the 32% in 2012, but a decline in growing stock does not necessarily mean a decline in forest area, for example thinning. Please rephrase.

   Response: We split the sentence into two to separate the different meanings.

3. L109 "The growing space "approximately corresponds to the crown projection area"(Riedel et al., 2017, pp. 39, author's translation), so we use these terms interchangeably in the following" No this should not be used interchangeably because it is very confusing to the reader. "approximately corresponds" is not sufficient to use the terms interchangeably. The "growing space" here is not defined, neither are its units. Is it cubic meter? This should really be changed, be careful with the use of terms and their units.

   Response: We added a sentence defining the growing space. Additionally, we clarified that the growing space and the crown area are not identical, and that we use the growing space as a proxy for the crown area. In the absence of a model for its direct estimation, it is the best approximation. The remainder of the text uses the term "crown area".

4. L113-L114 still very unclearly written, not sure how trees were selected as visible. Please rephrase.

   Response: We rephrased the passage again and hope that it is clearer now.

5. L196 "Obviously, broadleaf trees exhibit a much stronger seasonal pattern", not if they are evergreen broadleaf trees. . . please separate leaf shape (broadleaf or needleleaf) and phenology (evergreen and deciduous) more clearly. For example, holly (Ilex) is an evergreen broadleaf tree/shrub in Europe that does not likely show an obvious seasonal pattern in reflectance. On the other hand, Larch (Larix) is a common deciduous coniferous needleleaf tree. Suggested edit: "Obviously, deciduous broadleaf trees exhibit a much stronger seasonal pattern than the evergreen coniferous trees in our dataset."

   Response: Changed as suggested, sorry that this issue appeared again.

**2 Reviewer 2**

**2.1 Major comments**

1. The authors claim that the reason for assigning pixel spectra to single trees instead of tree composition parameters is the inaccuracy of tree locations derived from Bitterlich sampling. First, most/all large area studies that classify tree species suffer from the same inaccuracies. Yet, they train and validate their models the way I described. Second, if the field data lacks precision then the reference data should reflect that, i.e., a Sentinel-2 pixel corresponds to mixtures of trees and not individual trees.

   Response: The inaccuracies of tree locations do not arise from the Bitterlich sampling method itself. They mainly stem from imprecise GNSS measurements of the plot center, which is used as the reference point for tree locations. We believe this is sufficiently explained in Section 2.5. Many studies that use field reference data at the individual tree level face similar challenges. However, we would like to clarify that the practices for obtaining species labels are not as standardized as claimed. We have now included a brief literature overview of the different approaches (lines 127-131). The publications we reviewed all employed different methods for generating training labels from field data. Some focused on dominant species only, while others considered species shares. Some used forest inventory points, while others relied on polygons derived from forest management data, which have their own limitations. Additionally, some studies sample individual pixels within polygons, others use fixed-area plots, or calculate reflectances at the level of individual tree crowns, similar to our approach. Given these variations, our choice of extracting reflectances from individual trees is just one of many possible options.

2. The authors write I had suggested to train a classifier that predicts a certain tree species composition, whereas their approach yields probabilities of tree species occurrence. I must clarify that labeling pixels according to their tree species composition was merely one example how the authors could retain the complexity of the inventory data. Another way would be to assign to each pixel a tree list or estimates of species-specific basal area or cover (from their estimated crown area). Most importantly, predicting a discrete class or a probability estimate is besides the point. The training data I suggested can also be used to predict tree species probabilities. The argument is about the support size and labels of the reference observations and not the choice of estimators or prediction algorithm. Important for the argument is, that you claim to have produced a reflectance database for individual trees using up to 20x20 m pixel sizes, whereas the reflectances at that scale are a mix of trees, tree species, and background reflectance. In homogenous, single species stands, this may not matter as much, though I will later make the argument that it is still better to label pixels instead of trees in that case. Mixed species stands are often eliminated from training data because the inventory data is not precise enough to link species to pixels in those instances. Your dataset sill contains mixed species pixels but labels them according to single species. From the perspective of model training, this introduces unnecessary noise.

   Response: We argue that there are basically two approaches for linking field and satellite data, which we now introduce in section 2.4. as "tree-centric" and "pixel-centric" (lines 133-137). The tree-centric approach we chose aims to extract the most probable reflectance values for a given tree crown, while the pixel-centric approach attempts to label a set of pixels using metrics derived from field data, such as the ones suggested by the reviewer. We openly discuss the disadvantages of the tree-centric approach in section 4.5 and address the related geolocation errors in section 4.1. To provide just two examples of why the pixel-centric approach is not necessarily superior to the tree-centric approach: first, many studies work with pure stands based on an arbitrary definition of purity. Some classify stands as pure when the majority species has a share of more than 50% (Xi et al. (2021): `https://ieeexplore.ieee.org/document/9495140`), while others require more than 80% (Verhulst et al. (2024): `https://www.mdpi.com/2072-4292/16/14/2653`). Second, in the case of Bitterlich sampling, the support area is undefined and must be estimated, e.g. based on the tree diameters and the basal area factor (Blickensdörfer et al. (2024): `https://www.sciencedirect.com/science/article/pii/S0034425724000804`). We argue that is a priori unclear whether the errors introduced by these simplifications and assumptions are smaller than the errors occurring for the tree-centric approach. This is a research question that will certainly be addressed in the future.

3. The effect of oversampling field plots: This is a data publication and not a scientific article with scientific hypothesis. So as a reviewer, I try to consider the consequences for other users, i.e., can the dataset be used for the intended purpose and is the application and its limitations clear? If a dataset is incorrectly used, it can do more damage than good to the community. So, what is the purpose of this dataset? Training a foundation model? Producing maps? Both are fine but come with different requirements. When producing maps, we want to use the estimated model errors to infer map errors, which requires a probability sample. For land cover mapping studies, it is not as big of an issue to separate training datasets from validation datasets. Here, reference land cover data is relatively easy to come by. However, reference data for mapping tree species can only be obtained in-situ. In the case of the presented training dataset, any model errors estimated from boostrapping or cross-validation will be difficult to interpret because of the oversampling of the inventory plot. Although the NFI subplots follow a probability sample, the created training data focusing on individual trees is not. As such, such errors are not unbiased estimates of map accuracy (unlike those reported in previous studies). Now, it is possible to put the responsibility to the map users, but due to the lack of reference data, it is unlikely that users will follow good mapping principles or know how to. At the very least, the data publication should make a recommendation or be clear about what user's shouldn't do.

   Response: We believe that the community is well-equipped to handle diverse datasets and to advance to a new state of the art, should someone release a dataset better suited for the given task. To clarify the intended use case of the dataset and its limitations, we have added a sentence to the abstract and two sentences to the introduction (lines 67-69). Additionally, we have introduced a new discussion section, 4.7, titled 'Considerations for Map Producers' (lines 307 ff), that addresses the "Dos" and "Don'ts". In this section, we emphasize that users should be cautious when judging map accuracy based on model accuracy. Instead, we recommend using zonal statistics, such as comparing species distributions (of visible trees) at the state level or against published estimates of tree species abundance from the NFI. Finally, we suggest using auxiliary data for definitive validation.

4. It is reasonable to request that the authors provide sufficient information on how to use or not use this dataset, particularly as the dataset does not follow standard best practices. I am not suggesting, that the authors envision all potential use cases, but it would be good to understand and communicate the

limitations of the data. In this regard, I would ask the authors to add the information about the effect of added noise on model accuracies in the text so that it is citable.

Response: The newly introduced section 4.7 addresses the limitations of the data. However, we chose not to include a new passage describing the effects of added noise on model accuracies. Such a discussion would necessitate a detailed explanation of the methods used to assess this effect, which falls outside the scope of this work.

**2.2 Minor comments**

1. L135: What are these new possibilities?

Response: We removed the text passage in this position and now list the possibilities in section 4.5 (line 280). The new possibility is mainly that it allows to extract data for rare species and such, that only appear in mixed stands, albeit their statistics will be influenced by mixed pixels.

2. Response to comment 15: To be correct, the geolocation error should be below half the size of a pixel. Since the error estimate specifies a range, +/- 9.5 m are OK for a pixel size of 20 m not 10 m. Also, the GNSS error of 5 m of the forest inventory data is surprisingly low. Can you provide more information how you obtain this estimate or a publication? Does this estimate only apply to the differentially corrected plots?

Response: We added a text passage to section 4.1, that states why the geolocation error of 9.4 m is likely an overestimation. Regarding the GNSS errors: these errors were determined by us, as described in section 2.5 and 3.5. We analyzed the GNSS errors of the NFI measurements by matching tree positions to ortho images and Figure 13 shows the result (11.2m at 95% confidence / 5m at 81% confidence), which does not significantly differ between corrected and uncorrected measurements. Generally, the GNSS measurements are averaged positions of 100 measurements over 100 seconds. In addition, 76.5% of the measurements were corrected differentially using terrestrial reference stations. The dataset includes the correction status, so that users can filter by this property.

**3 Reviewer 3**

**3.1 Major comments**

1. The authors cover many important topics regarding reference data for large area mapping of tree species and supply a valuable dataset for the research community. While the dataset will lack behind expectation with regard to scientific freedom and flexibility, its provision will provide researchers, educators and students with additional data for the investigation of important research questions related to climate change, forest preservation, forest management and biodiversity studies. While some aspects of the data set, such as preprocessing, could be up for discussion, its publication will serve as a baseline for future publications of state and federal data sets and hopefully motivate more government authorities to provide their inventory data to the public. The writing style is excellent throughout the paper with only a single recommendation from my side. The researchers worked thoroughly on assuring high data quality and investigated the data set at hand for important characteristics, such as distinguishability of species from spectral signatures and geolocation accuracy, something that is to be expected from future related research. In my opinion, a few aspects of possible data usage were missed in the study design and discussion but overall, the state of the art in the field of tree species mapping with multi spectral satellite imagery is presented correctly.

Response: We thank the reviewer for the positive comment. We now give a hint about possible data usages beyond machine learning in the introduction (line 68).

**3.2 Minor comments**

1. 106-107: "if their crown is overlapped by less than 50% by the surrounding trees" would be clearer language IMO

Response: We revised this passage and included your wording.

2. 2.3: Additional TSA-processing withing the FORCE framework is not possible and the opportunity to create a dataset of even higher quality is missed. This would undoubtedly lower the amount of available tree observations but might help classification approaches that are sensitive to noise from cloud shadows and fog.

Response: TSA processing requires deciding on many individual processing parameters, which would have to be tailored to specific user needs. Furthermore, it requires high amounts of processing time and disk space. In consequence, we decided against it, especially as nowadays classification methods exist that can work with non-equitemporal time series (transformer-style neural networks for example).

3. 134-135: calculating the area-weighted average of a pixels might be a big source of noise if, let's say, the other 75% of that pixel depict a substantially different type of land cover than the target tree species. Think of the spectral signature of a deciduous tree that is added to a coniferous evergreen and the undergrowth signal in winter observations. In my opinion, some sort of outlier detection should be put in lace to detect possible addition of noise.

Response: Due to the angle count sampling design, that does not measure all trees within a given area, we do not have complete information about a plot's coverage. In consequence, the described effect can occur and we discuss it in section 4.5. Computing the area-weighted average, however, reduces the noise, as it tries to be as precise as possible, even under uncertain conditions. We refrained from making further assumptions regarding which trees to include / exclude from the dataset, as we already filtered the trees based on their probable visibility. However, the end user of the dataset is free to implement further quality filters.

4. 276: This might be due to Pinus' often very top-heavy crown in plantations that allows undergrowth to be more visible. In combination with Betulas characteristic bark, it is no wonder that the signal gets mixed up. There might be similar issues with stands including Larix or Fraxinus. 310: One additional idea for use of your dataset could be the investigation of mixed pixels. For large area mapping it would be great to know if any given mixture of species within a single pixel can be learned by a classifier.

Response: We added a sentence explaining that these species combinations were chosen because they are often co-occurring (line 229).

5. 310: One additional idea for use of your dataset could be the investigation of mixed pixels. For large area mapping it would be great to know if any given mixture of species within a single pixel can be learned by a classifier.

Response: Unfortunately, this is not possible with the presented dataset, as it is impossible to derive a species share for a given area based on the included data. Section 4.5 (line 294 ff) lists the angle count sampling as underlying reason for this problem.

**3.3 Other comments**

While I see possible issues with area-weighted pixel extraction, I do not agree with the criticism stated by Reviewer 2: FORCE uses the ImproPhe algorithm (Frantz 2016) that alters the pixel values of the 20m bands. To my understanding, duplicate values within the vicinity of a datapoint and thus spatial autocorrelation will be quite unlikely given the large size of the dataset. I can also support the author's claim, that duplicates (as well as random noise) within a certain threshold as stated in regard to the LAION 400M dataset are no issues for modern machine learning algorithms, especially neural networks, from my personal experience. The addition of random noise is a valid point of criticism. However, as long as European NFI rely on fixed position sample plots, this approach seems to be the only viable method to provide data to the research community.